# Multimodal Quantitative Language for Generative Recommendation

**Jianyang Zhai[1,2], Zi-Feng Mai[1,3], Chang-Dong Wang[1,3]\*, Feidiao Yang[2]\*,**
**Xiawu Zheng[2,4], Hui Li[4], Yonghong Tian[2,5]**
[1]Sun Yat-sen University, [2]Pengcheng Laboratory,
[3]Guangdong Key Laboratory of Big Data Analysis and Processing,
[4]Xiamen University, [5]Peking University
{zhaijy01, yangfd}@pcl.ac.cn, changdongwang@hotmail.com

## Abstract

Generative recommendation has emerged as a promising paradigm aiming at directly generating the identifiers of the target candidates. Most existing methods attempt to leverage prior knowledge embedded in Pre-trained Language Models (PLMs) to improve the recommendation performance. However, they often fail to accommodate the differences between the general linguistic knowledge of PLMs and the specific needs of recommendation systems. Moreover, they rarely consider the complementary knowledge between the multimodal information of items, which represents the multi-faceted preferences of users. To facilitate efficient recommendation knowledge transfer, we propose a novel approach called Multimodal Quantitative Language for Generative Recommendation (MQL4GRec). Our key idea is to transform items from different domains and modalities into a unified language, which can serve as a bridge for transferring recommendation knowledge. Specifically, we first introduce quantitative translators to convert the text and image content of items from various domains into a new and concise language, known as quantitative language, with all items sharing the same vocabulary. Then, we design a series of quantitative language generation tasks to enrich quantitative language with semantic information and prior knowledge. Finally, we achieve the transfer of recommendation knowledge from different domains and modalities to the recommendation task through pre-training and fine-tuning. We evaluate the effectiveness of MQL4GRec through extensive experiments and comparisons with existing methods, achieving improvements over the baseline by 11.18%, 14.82%, and 7.95% on the NDCG metric across three different datasets, respectively. [1]

## 1 Introduction

Recommendation systems (RS) aim to recommend items to users that they may be interested in, and are widely used on many online platforms, such as e-commerce and social networking (Chaves et al., 2022; Covington et al., 2016). For a long time, recommendation models that represent users and items using their unique IDs (known as IDRec) have been dominant in the field of RS (Kang & McAuley, 2018a; Sun et al., 2019; Zhang et al., 2024). However, IDRec may encounter cold start and knowledge transferability issues due to its inherent properties. To address

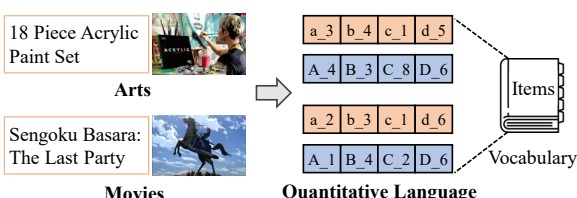

Figure 1: Illustration of our MQL4GRec. We translate items from different domains and modalities into a new unified language, which can then serve as a bridge for transferring recommendation knowledge.

---

\*Corresponding authors.
[1]Our implementation is available at: https://github.com/zhaijianyang/MQL4GRec.

these limitations, some literature (Hou et al., 2022; Sun et al., 2023) employs modal encoders (Devlin et al., 2018; He et al., 2016) to learn universal representations of items or sequences. While promising, these modal encoders are typically not specifically designed for recommendation tasks, resulting in suboptimal performance.

Recently, generative recommendation has emerged as a promising paradigm, which employs an end-to-end generative model to directly predict identifiers of target candidates (Geng et al., 2022; Rajput et al., 2023). Due to the success of PLMs in natural language generation (NLG) (Raffel et al., 2020a; Brown et al., 2020; Touvron et al., 2023), most existing methods attempt to leverage the prior knowledge of PLMs to improve the recommendation performance (Bao et al., 2023; Zhang et al., 2023; Zheng et al., 2023). They formalize the recommendation task as a sequence-to-sequence generation process, where the input sequence contains data of items interacted with users, and the output sequence represent identifiers of target items. Then they enable PLMs to perform recommendation tasks by adding instructions or prompts. Despite achieving decent performance, they suffer from the following limitations: 1) There are significant task differences between PLMs and RS, which may lead to inconsistencies between the general linguistic knowledge of PLMs and the specific requirements of RS; 2) They often overlook the complementary knowledge between the multimodal information of items, which is crucial for capturing the multi-faceted preferences of users.

To address these limitations, it is crucial to bridge the gaps between different domains and modalities, leveraging their recommendation knowledge to enhance the performance of the target domains. Inspired by significant advancements in NLG, such as pretraining-finetuning (Devlin et al., 2018; Raffel et al., 2020b) and prompt-tuning (Brown et al., 2020; Touvron et al., 2023), we propose the idea of transforming items from various domains and modalities into a new and unified language. A key factor contributing to these significant advances is the use of a shared vocabulary, where tokens are endowed with rich semantic information and prior knowledge across various tasks, which can then be effectively transferred to downstream tasks. Thus, we aspire for this new language to encompass a vocabulary in which tokens can represent items from various domains and modalities, as depicted in Figure 1. Specifically, this language not only serves as a bridge for knowledge transfer but also as identifiers of items, and should be more concise than the original modalities (text and image) to avoid issues in generation (Hua et al., 2023).

To this end, we propose a novel approach known as Quantitative Language for Multimodal Generative Recommendation (MQL4GRec). Specifically, we first introduce quantitative translators to convert the content of items (text and images) into the quantitative language. We train a separate quantitative translator for each modality of the item, each consisting of a modal encoder and a vector quantizer. Together, the codebooks of the two quantitative translators constitute the vocabulary. Then, we design a series of quantitative language generation tasks aiming at endowing quantitative language with rich semantic information and prior knowledge, and these tasks can be viewed as microcosms of NLG tasks. Specifically, we additionally incorporate some special tokens as task prompts. Finally, we transfer the source domain and multimodal recommendation knowledge to the recommendation tasks through pre-training and fine-tuning. To evaluate the effectiveness of our proposed MQL4GRec, we conduct extensive experiments and comparisons with existing methods. Relative to the baseline, we observe improvements of 11.18%, 14.82%, and 7.95% on the NDCG metric across three datasets, respectively. In summary, our proposed MQL4GRec achieves the transfer of recommendation knowledge by breaking down barriers between items across different domains and modalities, demonstrating strong scalability and potential. Our main contributions can be summarized as follows:

- We propose MQL4GRec, a novel approach that translates items from various domains and modalities into a unified quantitative language, thereby breaking down the barriers between them and facilitating the transfer of recommendation knowledge.

- We design a series of quantitative language generation tasks that endow quantitative language with rich semantic information and prior knowledge, and enhance the performance of recommendation tasks through pre-training and fine-tuning.

- We conduct extensive experiments and analyses on three public datasets, and the results validate the effectiveness of our proposed method.

## 2 RELATED WORKS

**Generative Recommendation.** Generative models are one of the hottest research topics in machine learning, resulting in some representative works such as Variational AutoEncoders (VAEs) (Kingma & Welling, 2014), Generative Adversarial Networks (GANs) (Goodfellow et al., 2014) and Diffusion models (Ho et al., 2020). Generally, generative models aim to learn the distribution of the training data $\mathbb{P}(\mathbf{x})$ and generate new samples $\mathbf{z} \sim \mathbb{P}(\mathbf{x})$. These generative models have also been applied to recommendation, resulting in many remarkable works of VAE-based (Cai & Cai, 2022; Shenbin et al., 2020), GAN-based (He et al., 2018; Guo et al., 2022; Wang et al., 2022) and diffusion-based (Jiang et al., 2024; Wang et al., 2023c) recommendation.

Recently, Transformer-based PLMs such as LLaMA (Touvron et al., 2023) and GPT (Brown et al., 2020) have also shown promising capabilities in language generation. With the help of such powerful generative PLMs, some PLM-based recommendation methods have also been proposed. Some early works, such as P5 (Geng et al., 2022) and M6-Rec (Cui et al., 2022), attempt to transform recommendation into a language generation task by designing prompts to bridge the gap between the downstream task and the pretraining task of PLMs. Some works focus on leveraging the prior knowledge in PLMs for recommendation by various tuning techniques such as parameter-efficient fine-tuning (PEFT) (Bao et al., 2023) and instruction tuning (Zhang et al., 2023).

One of the most important tasks in PLM-based recommendation is how to assign an unique sequence of tokens to each item as its ID. Early works (Geng et al., 2022; Cui et al., 2022) directly use the original name of the item or randomly assign an integer for each item, which have weak transferability and are sometimes unintelligible to PLMs. SEATER (Si et al., 2023) constructs tree-structured item IDs from a pretrained SASRec (Kang & McAuley, 2018b) model. P5-ID (Hua et al., 2023) investigates the effect of different item IDs on recommendation. ColaRec (Wang et al., 2024) captures the collaborative signals between items to construct generative item IDs. Notably, TIGER (Rajput et al., 2023) is the first attempt to use RQ-VAE to construct item IDs by quantizing the item embeddings.

**Multi-modal Recommendation.** Multi-modal side information of items, such as descriptive text and images, has been shown to be effective in improving recommendations by providing richer contexts for interactions. Early works such as VBPR (He & McAuley, 2016) extract visual features by matrix factorization to achieve more personalized ranking. Some works (Wei et al., 2019; Sun et al., 2020; Wei et al., 2020) leverage various types of graph neural network (GNN) to fuse the multi-modal features. For example, LATTICE (Zhang et al., 2021) designs a modality-aware learning layer to learn item-item structures for each modality and aggregates them to obtain latent item graphs. DualGNN (Wang et al., 2023b) proposes a multi-modal representation learning module to model the user attentions across modalities and inductively learn the user preference. MVGAE (Yi & Chen, 2022) uses a modality-specific variational graph autoencoder to fuse the modality-specific node embeddings.

Recently, with the profound development of foundation models in different modalities (Radford et al., 2021; Brown et al., 2020; Raffel et al., 2020b), some recent works attempt to leverage pretrained foundation models as feature encoders to encode the multi-modal side information. Following P5 (Geng et al., 2022), VIP5 (Geng et al., 2023b) extends it into a multi-modal version which encodes the item images by a pretrained CLIP image encoder. MMGRec (Liu et al., 2024) utilizes a Graph RQ-VAE to construct item IDs from both multi-modal and collaborative information. Moreover, IISAN (Fu et al., 2024) propose a simple plug-and-play architecture using a Decoupled PEFT structure and exploiting both intra- and inter-modal adaptation.

## 3 METHOD

In this section, we elaborate on the proposed MQL4GRec, a novel approach of transferring recommendation knowledge across different domains and modalities. We first translate item content into a unified quantitative language, which bridge the gaps between different domains and modalities. Then, we design a series of quantitative language generation tasks, and achieve the transfer of recommendation knowledge through pre-training and fine-tuning. The overall framework of the method is illustrated in Figure 2.

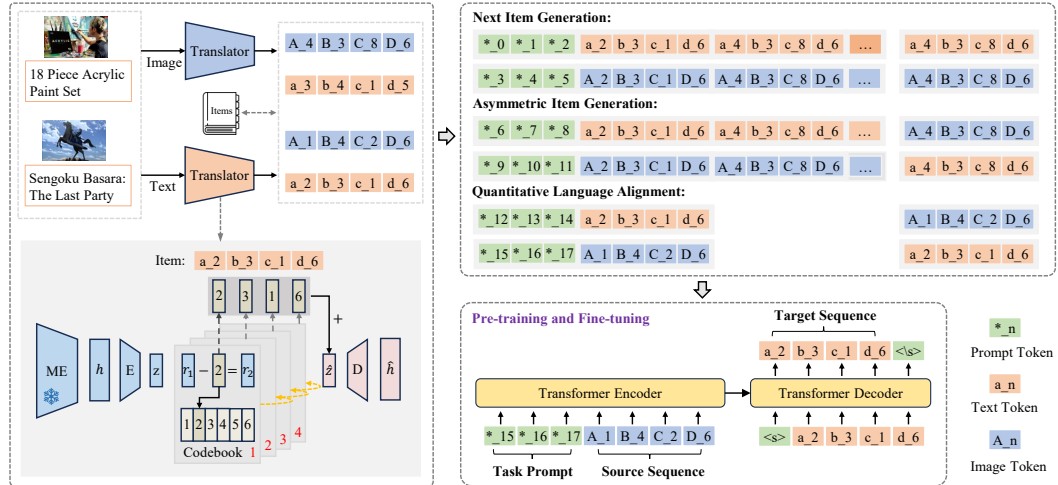

Figure 2: The overall framework of MQL4GRec. We regard the quantizer as a translator, converting item content from different domains and modalities into a unified quantitative language, thus bridging the gap between them (left). Subsequently, we design a series of quantitative language generation tasks to facilitate the transfer of recommendation knowledge through pre-training and fine-tuning (right).

## 3.1 QUANTITATIVE LANGUAGE

The original modal content of items is complex, which can affect the efficiency and performance of recommendations (Hua et al., 2023). Therefore, we translate item content from various domains and modalities into a concise and unified quantitative language. In this subsection, we introduce a quantitative translator to accomplish the aforementioned conversion.

**Quantitative Translator.** Vector Quantization (VQ) is an information compression technique widely utilized across various domains (Van Den Oord et al., 2017; Zeghidour et al., 2021), which maps high-dimensional data onto a finite set of discrete vectors, known as the codebook. In this paper, we treat the quantizer as a translator that converts complex item content into a concise quantitative language. Here, the codebook serves as the vocabulary of the quantitative language.

To obtain a unified quantitative language, we first employ a frozen modal encoder (LLaMA or ViT (Dosovitskiy et al., 2020)) to encode item content (text or image), and to obtain the item representation. Further, we take the item representation as input, and train a Residual-Quantized Variational AutoEncoder (RQ-VAE) (Zeghidour et al., 2021) for generating item tokens. RQ-VAE is a multi-level vector quantizer that applies quantization on residuals to generate a tuple of codewords (*i.e.*, item tokens). As shown in Figure 2 (left), for an item representation $\boldsymbol{h}$, RQ-VAE first encodes it into a latent representation $\boldsymbol{z}$. At each level $\boldsymbol{l}$, we have a codebook $\mathcal{C}^l = \{\boldsymbol{v}_k^l\}_{k=1}^K$, where each codebook vector is a learnable cluster center. The residual quantization process can be represented as:

$$c_i = \arg\min_k \left\| \boldsymbol{r}_i - \boldsymbol{v}_k^i \right\|_2^2, \tag{1}$$

$$\boldsymbol{r}_{i+1} = \boldsymbol{r}_i - \boldsymbol{v}_{c_i}^i, \tag{2}$$

where $c_i$ is the codeword of the $i$-th level, $\boldsymbol{r}_i$ is the residual vector of the $i$-th level, and $\boldsymbol{r}_1 = \boldsymbol{z}$. Assuming we have L-level codebooks, the quantization representation of $\boldsymbol{z}$ can be obtained according to $\hat{\boldsymbol{z}} = \sum_{i=1}^L \boldsymbol{v}_{c_i}^i$. Then $\hat{\boldsymbol{z}}$ will be used as decoder input to reconstruct the item representation $\boldsymbol{h}$. The loss function can be represented as:

$$\mathcal{L}_{\text{recon}} = \|\boldsymbol{h} - \hat{\boldsymbol{h}}\|_2^2, \tag{3}$$

$$\mathcal{L}_{\text{rqvae}} = \sum_{i=1}^H \left\| \text{sg}\left[\boldsymbol{r}_i\right] - \boldsymbol{v}_{c_i}^i \right\|_2^2 + \beta \left\| \boldsymbol{r}_i - \text{sg}\left[\boldsymbol{v}_{c_i}^i\right] \right\|_2^2, \tag{4}$$

$$\mathcal{L}(h) = \mathcal{L}_{\text{recon}} + \mathcal{L}_{\text{rqvae}}, \tag{5}$$

where $\hat{h}$ is the output of the decoder, sg[*] represents the stop-gradient operator, and $\beta$ is a loss coefficient. The overall loss is divided into two parts, $\mathcal{L}_{\text{recon}}$ is the reconstruction loss, and $\mathcal{L}_{\text{rqvae}}$ is the RQ loss used to minimize the distance between codebook vectors and residual vectors.

Items typically encompass content from multiple modalities, representing various aspects of user preferences. In our setup, each item comprises two modalities: text and image. We train a quantitative translator for each modality, then add prefixes to the codewords from each of the two codebooks to form a dictionary. Specifically, for the text quantitative translator, we prepend lowercase letter prefixes to the codewords to obtain $V_t = \{a\_1, b\_2, \ldots, d\_K\}$; for the image quantitative translator, we prepend uppercase letter prefixes to the codewords to obtain $V_v = \{A\_1, B\_2, \ldots, D\_K\}$. Here, $a/A$ represents the 1-th level codebook, $d/D$ represents the 4-th level codebook, etc. Subsequently, the dictionary can be represented as $V = \{V_t, V_v\}$. With each quantitative translator having $LK$ codewords, the size of our dictionary is $2LK$, enabling us to represent a total of $K^L$ items.

Once the quantitative translators are trained, we can directly use them to translate new items into quantitative language. For example, for the item text *"Sengoku Basara: The Last Party"*, after encoding it through the text encoder and RQ-VAE, we obtain a set of codewords (2, 3, 1, 6). Then, by appending lowercase letters before each number, we can get the text quantitative language of the item as *<a_2><b_3><c_1><d_6>*. Similarly, for the item's image, we can obtain its image quantitative language as *<A_1><B_4><C_2><D_6>*.

**Handling Collisions.** Translating item content into quantitative language may lead to item collisions, where multiple items possess the same tokens. To address this issue, some methods (Rajput et al., 2023; Hua et al., 2023) append an additional identifier after the item indices, which may introduce semantically unrelated distributions. LC-Rec (Zheng et al., 2023) introduces a uniform distribution constraint to prevent multiple items from clustering in the same leaf node. However, this method does not completely resolve collisions, such as when items have the same modality information or when the number of collisions exceeds the size of the last level codebook, which can lead to inflated performance metrics. (More discussion in Appendix E.1.)

To address the above issue, we reallocate tokens for colliding items based on the distance from the residual vector to the code vectors. Specifically, for $N$ colliding items, we first calculate the distances $\boldsymbol{D} \in \mathbb{R}^{N \times L \times K}$ between the residual vectors and the code vectors for each level based on $\boldsymbol{d}_k^i = \left\| \boldsymbol{r}_i - \boldsymbol{v}_k^i \right\|_2^2$, and sort the distances to obtain the indices $\boldsymbol{I} = \text{argsort}(\boldsymbol{D}, axis = 2) \in \mathbb{R}^{N \times L \times K}$. Then, we sort the colliding items based on their minimum distance to the code vectors of the last level, i.e., $(item_1, item_2, \ldots, item_N) = \text{sort}_{\min(\boldsymbol{d}^L)}(colliding\ items)$. Finally, we reallocate tokens for the sorted colliding items based on $\boldsymbol{I}$, following these principles: 1) Start from the last level to assign the nearest token to each item. If collisions occur, assign the next nearest token. 2) If there are insufficient tokens in the last level, for the remaining colliding items, reallocate tokens from the second last level based on distance, and then reallocate tokens from the last level. We repeat this process until all colliding items are handled.

## 3.2 QUANTITATIVE LANGUAGE GENERATION TASKS

In this subsection, we design several quantitative language generation tasks with the aim of imbuing quantitative language with more semantic information, thereby transferring prior knowledge to the target task, as illustrated in Figure 2 (right). Specifically, we additionally include some special tokens in the dictionary, which can serve as prompts to differentiate the types of tasks.

**Next Item Generation.** Since our primary goal is to predict the next item, the next item generation task is our main optimization objective. Specifically, each item contains both text and image modalities, so we have two subtasks: 1) Next Text Item Generation; 2) Next Image Item Generation. In this context, the input sequence is the item tokens sequence from the user interaction history, and the output sequence is the target item tokens corresponding to the respective modality. Different modal sequences reflect different aspects of user preferences.

**Asymmetric Item Generation.** In the next item generation task, the input and output are tokens of the same modality, and we refer to this task as symmetric. To facilitate the interaction

of recommendation knowledge between two modalities, we introduce asymmetric item generation tasks. Here, there are two subtasks: 1) Asymmetric Text Item Generation, where the input is the image tokens of the interaction history items, and the output is the text tokens of the target item; 2) Asymmetric Image Item Generation, where the input is the text tokens of the interaction history items, and the output is the image tokens of the target item. For example, for the input sequence *"<\*_6><\*_7><\*_8><a_2><b_3><c_1><d_6><a_4><b_3><c_8><d_6>"*, in human-understandable language, it can be described as follows: *"Based on the user's text interaction sequence, please predict the next item's image quantitative language: <a_2><b_3><c_1><d_6>, <a_4><b_3><c_8><d_6>"*.

**Quantitative Language Alignment**  Asymmetric item generation tasks enable the interaction of knowledge between two modalities, but they fall under the category of implicit alignment of the two modalities. We further introduce explicit Quantitative Language Alignment tasks to directly achieve alignment between the text and image quantitative languages of items. Here, we also have two subtasks: 1) Text-to-Image Alignment; 2) Image-to-Text Alignment. For example, for the input sequence *"<\*_12><\*_13><\*_14><a_2><b_3><c_1><d_6>"*, in human-understandable language, it can be described as follows: *"Please provide the image quantitative language for the following item: <a_2><b_3><c_1><d_6>"*.

### 3.3 TRAINING AND RECOMMENDATION

**Training.**  Quantitative language can be viewed as a microcosm of natural language. We employ a two-stage paradigm of pre-training and fine-tuning to optimize the model, which is similar to NLG tasks. For **pre-training**, we utilize the source domain datasets, where the pre-training task consists of two sub-tasks for next item generation. The purpose is to transfer recommendation knowledge from the source domains to the target domains. For **fine-tuning**, we conduct it on the target domain dataset, with tasks encompassing all quantitative language generation tasks. The aim is to leverage recommendation knowledge from different modalities to explore users' multifaceted preferences. The tasks mentioned above are conditional language generation tasks performed in a sequence-to-sequence manner. We optimize the negative log-likelihood of the generation target as follows:

$$\mathcal{L}_\theta = -\sum_{j=1}^{|\mathbf{Y}|} \log P_\theta \left( \mathbf{Y}_j \mid \mathbf{Y}_{<j}, \mathbf{X} \right), \tag{6}$$

where $\theta$ is the model parameters, $\mathbf{X}$ is the input sequence of encoder, and $\mathbf{Y_j}$ is the $j$-th token of $\mathbf{Y}$.

**Re-ranking for recommendation.**  There are two sub-tasks in the next item generation task, representing different user preferences. Although fine-tuning tasks can facilitate the transfer of recommendation knowledge between them, there might be some information loss. Therefore, we re-rank items by utilizing the recommendation lists generated from the two sub-tasks. The basic idea is that items appearing in both lists should be ranked higher. Specifically, we first obtain recommendation lists $R_t$ and $R_v$ for each sub-task through beam search, which include scores for each item. Then, the new score for each item can be formalized as:

$$s(x) = \begin{cases} (s_t(x) + s_v(x))/2 + 1 & x \in R_t, x \in R_v \\ s_t(x) & x \in R_t \\ s_v(x) & x \in R_v \end{cases}, \tag{7}$$

where $s_i(x)$ is the score of item $x$ in the list $R_i$, and $i \in \{t, v\}$.

## 4 EXPERIMENTS

### 4.1 EXPERIMENTAL SETTINGS

**Datasets.**  We evaluate the proposed approach on three public real-world benchmarks from the Amazon Product Reviews dataset (Ni et al., 2019), containing user reviews and item metadata from May 1996 to October 2018. In particular, we use six categories for pre-training, including *"Pet Supplies"*, *"Cell Phones and Accessories"*, *"Automotive"*, *"Tools and Home Improvement"*, *"Toys*

Table 1: Performance comparison of different methods on the three datasets. The best and second-best performances are indicated in bold and underlined font, respectively.

| Dataset | Metrics | GRU4Rec | BERT4Rec | SASRec | FDSA | S³-Rec | VQ-Rec | MISSRec | P5-CID | VIP5 | TIGER | MQL4GRec | Improv. |
|---------|---------|---------|----------|--------|------|--------|--------|---------|--------|------|-------|----------|---------|
| Instruments | HR@1 | 0.0566 | 0.0450 | 0.0318 | 0.0530 | 0.0339 | 0.0502 | 0.0723 | 0.0512 | 0.0737 | 0.0754 | **0.0833** | +10.48% |
| | HR@5 | 0.0975 | 0.0856 | 0.0946 | 0.0987 | 0.0937 | 0.1062 | 0.1089 | 0.0839 | 0.0892 | 0.1007 | **0.1115** | +2.39% |
| | HR@10 | 0.1207 | 0.1081 | 0.1233 | 0.1249 | 0.1123 | 0.1357 | 0.1361 | 0.1119 | 0.1071 | 0.1221 | **0.1375** | +1.03% |
| | NDCG@5 | 0.0783 | 0.0667 | 0.0654 | 0.0775 | 0.0693 | 0.0796 | 0.0797 | 0.0678 | 0.0815 | 0.0882 | **0.0977** | +10.77% |
| | NDCG@10 | 0.0857 | 0.0739 | 0.0746 | 0.0859 | 0.0743 | 0.0891 | 0.0880 | 0.0704 | 0.0872 | 0.0950 | **0.1060** | +11.58% |
| Arts | HR@1 | 0.0365 | 0.0289 | 0.0212 | 0.0380 | 0.0172 | 0.0408 | 0.0479 | 0.0421 | 0.0474 | 0.0532 | **0.0672** | +26.32% |
| | HR@5 | 0.0817 | 0.0697 | 0.0951 | 0.0832 | 0.0739 | **0.1038** | 0.1021 | 0.0713 | 0.0704 | 0.0894 | 0.1037 | - |
| | HR@10 | 0.1088 | 0.0922 | 0.1250 | 0.1190 | 0.1030 | **0.1386** | 0.1321 | 0.0994 | 0.0859 | 0.1167 | 0.1327 | - |
| | NDCG@5 | 0.0602 | 0.0502 | 0.0610 | 0.0583 | 0.0511 | 0.0699 | 0.0699 | 0.0607 | 0.0586 | 0.0732 | **0.0857** | +17.08% |
| | NDCG@10 | 0.0690 | 0.0575 | 0.0706 | 0.0695 | 0.0630 | 0.0844 | 0.0815 | 0.0662 | 0.0635 | 0.0806 | **0.0950** | +12.56% |
| Games | HR@1 | 0.0140 | 0.0115 | 0.0069 | 0.0163 | 0.0136 | 0.0075 | 0.0201 | 0.0169 | 0.0173 | 0.0166 | **0.0203** | +1.00% |
| | HR@5 | 0.0544 | 0.0426 | 0.0587 | 0.0614 | 0.0527 | 0.0408 | **0.0674** | 0.0532 | 0.0480 | 0.0523 | 0.0637 | - |
| | HR@10 | 0.0895 | 0.0725 | 0.0985 | 0.0988 | 0.0903 | 0.0679 | **0.1048** | 0.0824 | 0.0758 | 0.0857 | 0.1033 | - |
| | NDCG@5 | 0.0341 | 0.0270 | 0.0333 | 0.0389 | 0.0351 | 0.0242 | 0.0385 | 0.0331 | 0.0328 | 0.0345 | **0.0421** | +8.23% |
| | NDCG@10 | 0.0453 | 0.0366 | 0.0461 | 0.0509 | 0.0468 | 0.0329 | 0.0499 | 0.0454 | 0.0418 | 0.0453 | **0.0548** | +7.66% |

Table 2: Ablation study of handling collisions.

| Methods | Instruments | | Arts | | Games | |
|---------|-------------|--------|------|--------|-------|--------|
| | HR@10 | NDCG@10 | HR@10 | NDCG@10 | HR@10 | NDCG@10 |
| TIGER | 0.1221 | 0.0950 | **0.1167** | 0.0806 | 0.0857 | 0.0453 |
| TIGER w/o user | 0.1216 | 0.0958 | 0.1159 | 0.0810 | 0.0863 | 0.0464 |
| Handling Collisions | **0.1277** | **0.0987** | 0.1163 | **0.0844** | **0.0885** | **0.0473** |

*and Games"*, *"Sports and Outdoors"*, and three categories for sequential recommendation tasks, including *"Musical Instruments"*, *"Arts Crafts and Sewing"*, *"Video Games"*. We discuss the dataset statistics and pre-processing in Appendix A.

**Evaluation Metrics.** We use top-k Recall (Recall@K) and Normalized Discounted Cumulative Gain (NDCG@K) with K = 1, 5, 10 to evaluate the recommendation performance. Following previous works (Geng et al., 2022; Hua et al., 2023), we employ the *leave-one-out* strategy for evaluation. We perform full ranking evaluation over the entire item set instead of sample-based evaluation. For the generative methods based on beam search, the beam size is uniformly set to 20.

## 4.2 OVERALL PERFORMANCE

In this section, we compare our proposed approach for generative recommendation with the following sequential recommendation methods (which are described briefly in Appendix B): GRU4Rec (Hidasi et al., 2015), BERT4Rec (Sun et al., 2019), SASRec (Kang & McAuley, 2018b), FDSA (Zhang et al., 2019), S³-Rec (Zhou et al., 2020), VQ-Rec (Hou et al., 2023), MISSRec(Wang et al., 2023a), P5-CID (Hua et al., 2023), VIP5 (Geng et al., 2023a), and TIGER (Rajput et al., 2023). Results are shown in Table 1. Based on these results, we can find:

For non-generative recommendation methods, MISSRec often achieves better performance in most cases, demonstrating that introducing multimodal information of items can enhance recommendation performance. For generative baseline methods, VIP5 with image information does not achieve good results, which may be due to the modal differences between PLMs and image information. Furthermore, TIGER performs well on the Instruments and Arts datasets but does not exhibit superiority on the Games dataset. This may be due to TIGER's lack of auxiliary content information. In contrast, our proposed method introduces recommendation knowledge from different domains and modalities.

Compared to baseline methods, our proposed MQL4GRec achieves the best performance in most cases, especially with significant improvements on the NDCG metric. This superior performance can be attributed to two factors: 1) We translate item content from different domains and modalities into a unified quantitative language, breaking down barriers between them; 2) The series of QLG tasks we designed enable the transfer of recommendation knowledge to target tasks through pre-training and fine-tuning methods.

Table 3: Ablation study of various quantitative language generation tasks without pre-training.

| Modal | Tasks | Instruments | | Arts | | Games | |
|---|---|---|---|---|---|---|---|
| | | HR@10 | NDCG@10 | HR@10 | NDCG@10 | HR@10 | NDCG@10 |
| Text | $NIG_1$ | 0.1277 | 0.0987 | 0.1163 | 0.0844 | 0.0885 | 0.0473 |
| | NIG | 0.1275 | 0.0986 | 0.1205 | 0.0877 | 0.0928 | 0.0493 |
| | + AIG | 0.1279 | 0.0987 | 0.1249 | 0.0895 | 0.1002 | 0.0529 |
| | + QLA | **0.1282** | **0.0993** | **0.1293** | **0.0913** | **0.1010** | **0.0531** |
| Image | $NIG_2$ | 0.1243 | 0.0968 | 0.1117 | 0.0812 | 0.0881 | 0.0478 |
| | NIG | 0.1262 | 0.0986 | 0.1158 | 0.0848 | 0.0899 | 0.0487 |
| | + AIG | **0.1299** | 0.0998 | 0.1218 | 0.0878 | 0.1002 | 0.0534 |
| | + QLA | 0.1280 | **0.1001** | **0.1259** | **0.0901** | **0.1017** | **0.0540** |

## 4.3 ABLATION STUDY

**Handling Collisions.** We propose a method based on the distance between the residual vector and the codeword vector to resolve item collisions. To validate the effectiveness of our method, we compare it with the collision resolution approach in TIGER, which directly adds an item index layer to resolve item collisions, thereby introducing a semantically unrelated distribution. The experimental results are shown in Table 2. "TIGER w/o user" refers to the removal of the user ID token from the input sequence, which is also done to facilitate a fairer comparison. From the experimental results, it can be seen that our method of handling collisions is more rational and effective. Furthermore, results indicate that including a user ID token in the input degrades model performance, particularly on the Games dataset. We attribute this to TIGER representing tens of thousands of users with only 2000 tokens, leading to numerous user ID collisions.

**Quantitative language generation tasks.** We initially assess the effect of different QLG tasks on performance without the use of pre-training, and the results are shown in Table 3. (For more detailed results, please refer to Appendix D.1.) Various tasks include: (1) NIG: the next item generation task introduced in Section 3.2; (2) AIG: the asymmetric item generation task; (3) QLA: the quantitative language alignment task. In this list, tasks without subscripts indicate that two subtasks are used simultaneously. "Text" denotes evaluating performance by utilizing the next text item generation subtask (i.e., $NIG_1$); "Image" signifies evaluating performance by utilizing the next image item generation subtask (i.e., $NIG_2$).

The results indicate that several quantitative language generation tasks designed by us can significantly improve performance. Specifically, as the number of tasks increases, the performance of both $NIG_1$ and $NIG_2$ improves. This indicates that these tasks can enrich the quantitative language by incorporating semantic information and knowledge across different modalities. In summary, converting the multimodal content of items into a unified quantitative language effectively facilitates the transfer of recommendation knowledge.

**Pre-training.** We transfer recommendation knowledge from the source domain datasets to the target dataset through pre-training. Here, we employ $NIG_1$ to evaluate the recommendation performance, with the results shown in Table 4. (Additional results can be found in Appendix D.2.) QLG represents the quantitative language generation tasks without pre-training. Specifically, the pre-training task labeled *"$NIG_1$ w/ pre-training"* employs only $NIG_1$ from the source domain datasets. On the other hand, the *"QLG w/ pre-training"* task uses both $NIG_1$ and $NIG_2$.

The results indicate that, under a single modality, pre-training enhances the performance across three downstream datasets, demonstrating that prior knowledge from the source domain can be effectively transferred to downstream tasks. Under dual modalities, pre-training significantly improves performance on the Instruments and Arts datasets; however, it does not yield a notable improvement for the Games dataset, potentially due to overfitting. A more intuitive analysis of this phenomenon is provided in Section 4.4. Finally, we re-rank the items according to Equation (7) to generate the final recommendation list.

Table 4: Ablation study of pre-training and quantitative language generation tasks.

| Methods | Instruments | | Arts | | Games | |
|---|---|---|---|---|---|---|
| | HR@10 | NDCG@10 | HR@10 | NDCG@10 | HR@10 | NDCG@10 |
| (0) $NIG_1$ | 0.1277 | 0.0987 | 0.1163 | 0.0844 | 0.0885 | 0.0473 |
| (1) QLG | 0.1282 | 0.0993 | 0.1293 | 0.0913 | 0.1010 | 0.0531 |
| (2) $NIG_1$ w/ pre-training | 0.1334 | 0.1043 | 0.1305 | **0.0959** | 0.0950 | 0.0508 |
| (3) QLG w/ pre-training | 0.1362 | 0.1051 | 0.1314 | 0.0944 | 0.0995 | 0.0521 |
| (4) MQL4GRec ((3) + re-ranking) | **0.1375** | **0.1060** | **0.1327** | 0.0950 | **0.1033** | **0.0548** |

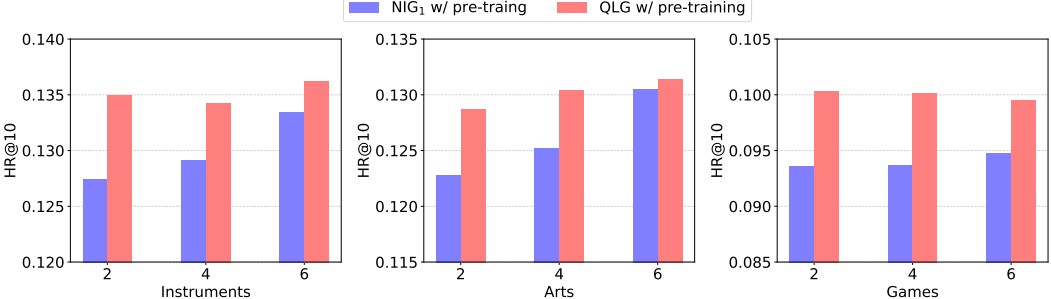

Figure 3: The impact of varying amounts of pre-training datasets on recommendation performance.

## 4.4 FURTHER ANALYSIS

**Pre-training datasets.** In this subsection, we investigate the impact of varying amounts of pre-training datasets on downstream tasks, and the results are shown in Figure 3. From the results, it can be observed that: 1) Exclusively in the context of text quantitative language, as the number of pre-training datasets increases, the performance of downstream tasks also gradually improves. This suggests that larger numbers in pre-training datasets provide more transferable recommendation knowledge. 2) Following pre-training with quantitative language under two modalities, fine-tuning shows varying trends across different downstream datasets. Specifically, while increasing pre-training datasets enhances performance on the Instruments and Arts datasets, it leads to a gradual decline in performance on the Games dataset. This could indicate either overfitting or significant domain differences between the Games dataset and the source domain datasets.

**Pre-training epochs** In this subsection, we investigate the impact of varying the number of pre-training epochs on downstream tasks, with the results displayed in Figure 4. From the figure, it can be observed that: 1) When pre-training is performed solely with text quantitative language, the performance of downstream tasks gradually increases with the number of pre-training epochs and stabilizes around 25 epochs. 2) When pre-training involves both text and image quantitative languages, the Instruments and Arts datasets reach peak performance early, and further training may impair the transfer of recommendation knowledge. In contrast, for the Games dataset, performance deteriorates as the number of pre-training epochs increases. This suggests that for the Games dataset, recommendation knowledge from different modalities might be more crucial than that from the source domain dataset, and there may be conflicts between the two.

Table 5: Zero-shot capabilities under different number of pre-training datasets.

| Number | Instruments | | Arts | | Games | |
|---|---|---|---|---|---|---|
| | HR@10 | NDCG@10 | HR@10 | NDCG@10 | HR@10 | NDCG@10 |
| 0 | 0.00099 | 0.00046 | 0.00113 | 0.00052 | 0.00066 | 0.00031 |
| 2 | 0.00240 | 0.00137 | 0.00140 | 0.00063 | 0.00109 | **0.00058** |
| 4 | 0.00310 | 0.00170 | 0.00298 | 0.00132 | 0.00054 | 0.00027 |
| 6 | **0.00345** | **0.00171** | **0.00311** | **0.00138** | **0.00116** | 0.00047 |

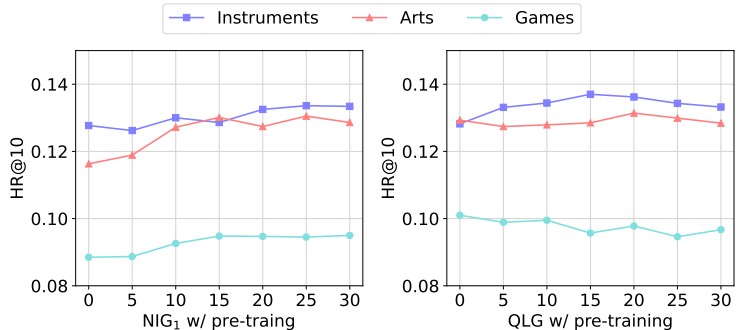

Figure 4: The impact of different pre-training epochs on recommendation performance.

### 4.5 ZERO-SHOT CAPABILITY

We investigate whether models pre-trained on the source domain dataset have zero-shot capabilities, as shown in Table 5. Here, "Number" represents the number of pre-training datasets, with "0" indicating model parameters randomly initialized. We use $NIG_1$ to evaluate performance. The results demonstrate that pre-trained models exhibit preliminary zero-shot capabilities on the Instruments and Arts datasets, although they are still weak. However, this capability is not evident on the Games dataset. We attribute this primarily to the scarcity of pre-training data and the limited parameters of the model, resulting in insufficient generalization. In the future, we aim to delve deeper into this phenomenon.

## 5 CONCLUSION

In this paper, we propose a novel approach named MQL4GRec, which transforms item content from different domains and modalities into a unified quantitative language to facilitate the effective transfer of recommendation knowledge. We first train a quantitative translator for each modality, converting items into the quantitative language and breaking down the barriers between them. Then, we design a series of quantitative language generation tasks aiming at endowing quantitative language with rich semantic information and prior knowledge. Finally, we transfer the source domain and multimodal recommendation knowledge to the recommendation tasks through pre-training and fine-tuning. Our proposed MQL4GRec achieves superior performance compared to the baseline method. Moreover, MQL4GRec possesses strong scalability and potential as it does not rely on traditional item IDs and bridges the gap between different domains and modalities. We believe this represents a significant step towards universal recommendation models.

ACKNOWLEDGMENTS

This work is supported by the National Natural Science Foundation of China (62206141, 62425101, 62332002, 62276277), the National Key Research and Development Program of China (2021YFF1201200), the Guangdong Basic and Applied Basic Research Foundation (2022B1515120059), and the Shenzhen Science and Technology Program (KQTD20240729102051063). In addition, we are grateful to Dr. Zhengyu Ma for her support.

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

## A    DATASET STATISTICS

Table 6: Statistics of the preprocessed datasets. "**Avg**. *len*" represents the average length of item sequences.

| Datasets | #Users | #Items | #Interactions | Sparsity | Avg. *len* |
|---|---|---|---|---|---|
| Pet | 183697 | 31986 | 1571284 | 99.97% | 8.55 |
| Cell | 123885 | 38298 | 873966 | 99.98% | 7.05 |
| Automotive | 105490 | 39537 | 845454 | 99.98% | 8.01 |
| Tools | 144326 | 41482 | 1153959 | 99.98% | 8.00 |
| Toys | 135748 | 47520 | 1158602 | 99.98% | 8.53 |
| Sports | 191920 | 56395 | 1504646 | 99.99% | 7.84 |
| Instruments | 17112 | 6250 | 136226 | 99.87% | 7.96 |
| Arts | 22171 | 9416 | 174079 | 99.92% | 7.85 |
| Games | 42259 | 13839 | 373514 | 99.94% | 8.84 |

To evaluate the performance of the proposed approach, we conduct experiments in the pre-trained source and target domain settings. We use six categories from from the Amazon Product Reviews dataset (Ni et al., 2019) for pre-training, including *"Pet Supplies"*, *"Cell Phones and Accessories"*, *"Automotive"*, *"Tools and Home Improvement"*, *"Toys and Games"*, *"Sports and Outdoors"*, and three categories for sequential recommendation tasks, including *"Musical Instruments"*, *"Arts Crafts and Sewing"*, *"Video Games"*.

Each item in the dataset is associated with a title, a description, and an image. Following previous work (Rajput et al., 2023), we first filter out unpopular users and items with less than five interactions. Then, we create user behavior sequences based on the chronological order. The maximum item sequence length is uniformly set to 20 to meet all baseline requirements. The statistics of our preprocessed datasets are shown in Table 6.

## B    BASELINES

We compare the proposed approach with the following baseline methods:

- **GRU4Rec** (Hidasi et al., 2015) introduces Gating Recurrent Unit (GRU) to model user action sequences for session-based recommendations.
- **SASRec** (Kang & McAuley, 2018b) uses a directional self-attentive model to capture item correlations within a sequence.
- **BERT4Rec** (Sun et al., 2019) employs a bi-directional self-attentive model with the cloze objective for modeling user behavior sequences.
- **FDSA** (Zhang et al., 2019) uses a self-attentive model to capture item and feature transition patterns.
- **S$^3$-Rec** (Zhou et al., 2020) pre-trains sequential models with mutual information maximization to learn the correlations among attributes, items, subsequences, and sequences.
- **VQ-Rec** (Hou et al., 2023) learns vector-quantized item representations for transferable sequential recommenders.
- **MISSRec** (Wang et al., 2023a) is a multi-modal pre-training and transfer learning framework for sequential recommendation.
- **P5-CID** (Geng et al., 2022; Hua et al., 2023) organizes multiple recommendation tasks in a text-to-text format and models different tasks uniformly using the T5 model. Here, we employ P5 with collaborative indexing as the baseline.
- **VIP5** (Geng et al., 2023a) is a multimodal foundation model considering visual, textual, and personalization modalities under the P5 recommendation paradigm, to unify various modalities and recommendation tasks.
- **TIGER** (Rajput et al., 2023) adopts the generative retrieval paradigm for sequential recommendation and introduces a semantic ID to uniquely identify items.

Table 7: Detailed ablation study of various quantitative language generation tasks without pre-training

| Modal | Tasks | Instruments | | Arts | | Games | |
|---|---|---|---|---|---|---|---|
| | | HR@10 | NDCG@10 | HR@10 | NDCG@10 | HR@10 | NDCG@10 |
| Text | $NIG_1$ | 0.1277 | 0.0987 | 0.1163 | 0.0844 | 0.0885 | 0.0473 |
| | $NIG_1$ + QLA | 0.1275 | 0.0986 | 0.1166 | 0.0831 | 0.0871 | 0.0465 |
| | NIG | 0.1275 | 0.0986 | 0.1205 | 0.0877 | 0.0928 | 0.0493 |
| | NIG + QLA | 0.1263 | 0.0983 | 0.1204 | 0.0867 | 0.0919 | 0.0492 |
| | NIG + AIG | 0.1279 | 0.0987 | 0.1249 | 0.0895 | 0.1002 | 0.0529 |
| | NIG + AIG + QLA | **0.1282** | **0.0993** | 0.1293 | 0.0913 | **0.1010** | **0.0531** |
| Image | $NIG_2$ | 0.1243 | 0.0968 | 0.1117 | 0.0812 | 0.0881 | 0.0478 |
| | $NIG_2$ + QLA | 0.1237 | 0.0978 | 0.1143 | 0.0826 | 0.0877 | 0.0468 |
| | NIG | 0.1262 | 0.0986 | 0.1158 | 0.0848 | 0.0899 | 0.0487 |
| | NIG + QLA | 0.1265 | 0.0988 | 0.1164 | 0.0849 | 0.0945 | 0.0505 |
| | NIG + AIG | **0.1299** | 0.0998 | 0.1218 | 0.0878 | 0.1002 | 0.0534 |
| | NIG + AIG + QLA | 0.1280 | **0.1001** | **0.1259** | **0.0901** | **0.1017** | **0.0540** |

## C IMPLEMENTATION DETAILS.

To obtain textual representations, we employ LLaMA to encode the title and description of the item as its embedding and use mean pooling to aggregate multiple representations. To obtain visual representations, we utilize CLIP's (Radford et al., 2021) image branch as an encoder to encode the images of items, and we employ ViT-L/14 as the backbone. Both the encoder and decoder of RQ-VAE are implemented as Multi-Layer Perceptrons (MLPs) with ReLU activation functions. The level of codebooks is set to 4, with each level consisting of 256 codebook vectors, and each vector has a dimension of 32. The model is optimized using the AdamW optimizer, employing a learning rate of 0.001 and a batch size of 1024.

Following previous work (Rajput et al., 2023), we use the T5 (Raffel et al., 2020b) framework to implement our transformer based encoder-decoder architecture. We use 4 layers each for the transformer-based encoder and decoder models with 6 self-attention heads of dimension 64 in each layer. The MLP and the input dimension was set as 1024 and 128, respectively. The number of prompt tokens for every task is set to 4. We employ the AdamW (Loshchilov & Hutter, 2019) optimizer for model optimization, setting the weight decay to 0.01. During pre-training, we utilize a batch size of 4096 with a learning rate set to 0.001. For alignment tuning, we employ a batch size of 512 with a maximum learning rate of 5e-4, and utilize a cosine scheduler with warm-up to adjust the learning rate.

Our experiments utilize the Tesla V100 GPU. For pretraining, we use four cards, and for fine-tuning, we use two cards. Since the model has around 13 million parameters, there is still a substantial amount of GPU memory remaining.

## D MORE ABLATION STUDIES

### D.1 QUANTITATIVE LANGUAGE GENERATION TASKS.

We have supplemented Table 7 with more detailed ablation experiments of quantitative language generation tasks without pre-training. We study the impact of each task on recommendation performance through a combination of different tasks. We find that: 1) the AIG task always results in a significant performance improvement; 2) the QLA task needs to be paired with the AIG task in order to achieve better results. This suggests that quantitative language serves as a bridge for knowledge transfer in recommendations, but we need to design appropriate quantitative language tasks.

### D.2 PRE-TRAINING.

We further provide the results of using $NIG_2$ to evaluate the performance of recommendations in Table 8. From the results, it can be seen that: 1) On the Instruments and Arts datasets, both pre-training and Quantitative Language Generation (QLG) tasks are useful for improving recommendation performance. This indicates that quantitative language can migrate recommendation knowledge from

Table 8: Detailed ablation study of pre-training and quantitative language generation tasks.

| Modal | Methods | Instruments | | Arts | | Games | |
|---|---|---|---|---|---|---|---|
| | | HR@10 | NDCG@10 | HR@10 | NDCG@10 | HR@10 | NDCG@10 |
| Text | (0) $NIG_1$ | 0.1277 | 0.0987 | 0.1163 | 0.0844 | 0.0885 | 0.0473 |
| | (1) QLG | 0.1282 | 0.0993 | 0.1293 | 0.0913 | 0.1010 | 0.0531 |
| | (2) $NIG_1$ w/ pre-training | 0.1334 | 0.1043 | 0.1305 | **0.0959** | 0.0950 | 0.0508 |
| | (3) QLG w/ pre-training | 0.1362 | 0.1051 | 0.1314 | 0.0944 | 0.0995 | 0.0521 |
| Image | (4) $NIG_2$ | 0.1243 | 0.0968 | 0.1117 | 0.0812 | 0.0881 | 0.0478 |
| | (5) QLG | 0.1280 | 0.1001 | 0.1259 | 0.0901 | 0.1017 | 0.0540 |
| | (6) QLG w/ pre-training | 0.1322 | 0.1029 | 0.1265 | 0.0914 | 0.0987 | 0.0526 |
| All | (7) MQL4GRec | **0.1375** | **0.1060** | **0.1327** | 0.0950 | **0.1033** | **0.0548** |

the source domain and other modalities to the target task. 2) On the Games dataset, QLG with pre-training impairs performance, which might be due to some conflict of recommendation knowledge between the source domain and another modality. In the future, we will further explore this phenomenon.

## E    DISCUSSION

### E.1    HANDLING COLLISIONS.

To address the issue of item collisions, some methods (Rajput et al., 2023; Hua et al., 2023) append an additional identifier to the item indices, which may introduce semantically unrelated distributions. LC-Rec (Zheng et al., 2023) introduces a uniform distribution constraint to prevent multiple items from clustering in the same leaf node. Although the LC-Rec achieves better performance in handling collisions compared to previous approaches, it has an inherent problem: it cannot completely resolve item collisions when item modal content is identical or when the number of collisions exceeds the size of the last level's codebook. This leads to another issue: **multiple items sharing the same indices results in an unfair comparison of performance**.

In contrast, our method of dealing with collisions is more rational and can essentially solve the aforementioned problems. From the experimental results, our approach achieved similar outcomes to LC-Rec, hence we did not use a dedicated table to list the experimental results.

### E.2    LIMITATIONS.

Although our method achieves state-of-the-art performance, there are still some inherent limitations. For example: 1) The inference time is longer compared to traditional recommendation methods. This is an inherent flaw of generative recommendation systems, as such methods typically employ beam search and auto-regressive techniques to generate the next token. 2) Our method requires item content information, and the scenario where item content is missing has not yet been studied in the paper. This is an issue that we need to further analyze in our next steps.

