# OpenReview forum: "Multimodal Quantitative Language for Generative Recommendation"
_ICLR.cc/2025/Conference — ICLR 2025 Poster_

### Official Review · Reviewer_rVKM · 2024-10-29

**Soundness:** 2
**Presentation:** 4
**Contribution:** 4
**Rating:** 8
**Confidence:** 4

**Summary:**

This paper introduces a novel approach, MQL4GRec, designed to convert item content from diverse domains and modalities into a unified quantitative language.

**Strengths:**

1. The proposed concept of a multimodal quantitative language, together with the design of quantitative language generation tasks, represents a novel and innovative advancement in the field of generative recommendation

2.  The architecture design is elegant, presenting the idea in a straightforward way, yet experiments demonstrate its strong effectiveness. I personally appreciate this type of work and believe it can make a meaningful impact in the field of generative recommendation.

3. The availability of the code significantly enhances reproducibility.

**Weaknesses:**

1.The proposed framework is rooted in the generative recommendation paradigm and aligns with a preprint in a similar research direction [1]. However, it still represents a valuable contribution in my view, even if not groundbreaking.

2.The authors should consider testing the statistical significance of MQL4GRec results.

3.There remains a limitation in zero-shot capability, which is a known challenge in the field of recommendation.

4.To enhance the comprehensiveness of the "Multi-modal Recommendation" section in the related work, the authors could consider including more recent state-of-the-art multimodal recommender system papers, such as [2,3]. The field of multimodal codebooks from other communities should also be included in the related work section to clarify the distinctions between the proposed MQL approach and existing methods, such as those in [4, 5]


[1] Liu, Han, et al. "MMGRec: Multimodal Generative Recommendation with Transformer Model." arXiv preprint arXiv:2404.16555 (2024).

[2] Fu, Junchen, et al. "IISAN: Efficiently adapting multimodal representation for sequential recommendation with decoupled PEFT." *Proceedings of the 47th International ACM SIGIR Conference on Research and Development in Information Retrieval*. 2024.

[3] Liu, Han, et al. "MMGRec: Multimodal Generative Recommendation with Transformer Model." *arXiv preprint arXiv:2404.16555* (2024).

[4] Lan, Zhibin, et al. "Exploring better text image translation with multimodal codebook." arXiv preprint arXiv:2305.17415 (2023).

[5] Duan, Jiali, et al. "Multi-modal alignment using representation codebook." Proceedings of the IEEE/CVF Conference on Computer Vision and Pattern Recognition. 2022.

**Questions:**

1.As an efficient framework, I am interested in understanding the training costs of MQL4GRec compared to other baselines. Specifically, how does it perform in terms of training time, VRAM usage, and inference time?

2.In the field of recommender systems, there is a lack of widely recognized pre-trained models. Could MQL4GRec potentially serve as a foundation model for other downstream tasks?

My concerns are addressed by the authors, therefore, I will raise my scores from 6 to 8.

---

> ### Author Response · Authors · 2024-11-22
> **Rebuttal by Authors**
>
> **Thank you very much for your insightful comments on our work！ We try to answer your questions as follows.**
>
> >*Q1: The authors should consider testing the statistical significance of MQL4GRec results.*
>
> A1: We conduct one-sample t-tests, and the results indicate that the improvements of MQL4GRec on the NDCG metric are statistically significant (p-value < 0.05).
>
> |                | Instruments  |         |        |        |         | Arts   |         |        |        |         | Games   |         |        |        |         |
> | ---------      | ------       | ------- | ------ | ------ |---------| ------ | ------- | ------ | ------ | ------  | ------  | ------- | ------ | ------ | ------  |
> |                | HR@1         | HR@5    | HR@10  | NDCG@5 | NDCG@10 | HR@1   | HR@5    | HR@10  | NDCG@5 | NDCG@10 |  HR@1   | HR@5    | HR@10  | NDCG@5 | NDCG@10 |
> | QL4GRec        | 0.0833       | 0.1115  | 0.1375 | 0.0977 |  0.1060 | 0.0672 | 0.1037  | 0.1327 | 0.0857 |  0.0950 |  0.0203 | 0.0637  | 0.1033 | 0.0421 |  0.0548 |
> | Improv.        | +10.48%      | +2.39%  | +1.03% | +10.77%|  +11.58%| +26.32%| -       | -      | +17.08%|  +12.56%|  +1.00% | -       | -      | +8.23% |  +7.66% |
> | p-value        | 1.74e-4      | 2.79e-1 | 5.93e-1| 1.32e-5|  5.04e-7|8.55e-17| 9.42e-1 | 1.02e-2|2.08e-12|  3.15e-9|  7.40e-1| 1.74e-3 | 3.09e-1| 1.48e-4|  8.80e-6|
>
> >*Q2: To enhance the comprehensiveness of the "Multi-modal Recommendation" section in the related work, the authors could consider including more recent state-of-the-art multimodal recommender system papers, such as [2,3]. The field of multimodal codebooks from other communities should also be included in the related work section to clarify the distinctions between the proposed MQL approach and existing methods, such as those in [4, 5]。*
>
> A2: Thank you very much for your suggestions. We will read the relevant papers and incorporate them into the related work section of the next version of the paper as soon as possible.

---

> > ### Author Response · Authors · 2024-11-22
> > **Rebuttal by Authors**
> >
> > >*Q3: As an efficient framework, I am interested in understanding the training costs of MQL4GRec compared to other baselines. Specifically, how does it perform in terms of training time, VRAM usage, and inference time?*
> >
> > A3: Our approach encompasses the training of vector quantization models, handling collisions, pre-training, fine-tuning, and inference stages:
> >
> > - The encoder and decoder of our vector quantization model are both composed of multi-layer perceptrons. The specific training time is related to the number of items. We set the batch size to 1024 and the maximum number of epochs to 500, using a Tesla V100 for training. Training the vector quantization model on six pre-training datasets takes approximately 30 minutes, occupying about 1.5GB of GPU memory.
> > - After training the vector quantization model, we assign tokens to each item and resolve item collisions. We perform one inference to obtain the tokens for each item and save the distance of the residual vectors to the codebook, then handle item collisions as described in the paper. This process takes only about 10 seconds.
> > - During the pre-training phase, we set the batch size per GPU to 1024 and the maximum number of epochs to 30, using four Tesla V100s. It takes approximately 10 hours on six pre-training datasets, occupying about 4*13.8GB of GPU memory. However, due to server issues, the GPU utilization is low, which is something that needs to be improved.
> > - In the fine-tuning stage, we set the batch size per GPU to 256, the maximum number of epochs to 200, and the early stop epochs to 10, using two Tesla V100s. Taking the Instruments dataset as an example, fine-tuning takes about 1.5 hours, occupying about 2*4.6GB of GPU memory.
> > - In the inference stage, we set the batch size per GPU to 64 and the beam size to 20, using two Tesla V100s. Taking the Instruments dataset as an example, inference takes about 7 minutes, occupying about 2*4.9GB of GPU memory.
> > - Fortunately, the vector quantization model and the pre-trained model only need to be trained once and can then be applied to multiple downstream datasets.
> >
> > We list the training time, VRAM usage, and inference time of different methods on the Instruments dataset in the table below. From the table, it can be observed that:
> >
> > - The inference time for non-generative methods is very short, as the number of items in the dataset is minimal. Generative methods require autoregressive decoding and beam search during inference, which often consumes more time.
> > - P5-CID and VIP5 are generative methods based on pre-trained language models. Due to the large number of model parameters, they require longer training and inference times.
> >
> > |                | GRU4Rec      | BERT4Rec|SASRec  |   FDSA |$S^3$-Rec| VQ-Rec |MISSRec  | P5-CID | VIP5   | TIGER   | MQL4GRec|
> > | ---------      | ------       | ------- | ------ | ------ |---------| ------ | ------- | ------ | ------ | ------  | ------  |
> > |Training time   | 5min         | 40min   | 5min   | 1day   | 3days   | 22min  | 30min   | 10h    | 10h    | 1.2h    |  1.5h   |
> > | VRAM           | 3.8GB        | 5.5GB   | 2.9GB  | 5.6GB  |  9.7GB  | 12.3GB | 2.5GB   | 2*20GB | 2*20GB | 2*4.6GB |  2*4.6GB|
> > | Inference time | <1min        | <1min   | <1min  | <1min  |  <1min  | <1min  | <1min   | 25min  |  25min | 7min    |  7min   |
> >
> > **Q4: In the field of recommender systems, there is a lack of widely recognized pre-trained models. Could MQL4GRec potentially serve as a foundation model for other downstream tasks?**
> >
> > A4: Personally, we believe that MQL4GRec has the potential to become a foundational recommendation model and provides a feasible approach for training such models, but it is still not ready to serve as a foundation model for other downstream tasks.
> >
> > - Firstly, as the results in Table 5 show, although the zero-shot capability improves with the increase of the pre-training dataset, its performance remains low.
> > - Secondly, MQL4GRec has a limited pre-training dataset, which may lead to significant domain differences when applied to downstream tasks.
> > - Lastly, the model parameters of MQL4GRec are quite small, with less than 10M, making it difficult for such a small-parameter model to become a foundational model.
> >
> > Foundational models have achieved significant success in the NLP field and have also made major breakthroughs in the CV field [1]. Particularly, the breakthroughs in the CV field offer many lessons that can be learned from. In the future, we will continue to delve into foundational models in the recommendation domain and explore the scaling laws of MQL4GRec.
> >
> >
> > [1] Bai Y, Geng X, Mangalam K, et al. Sequential modeling enables scalable learning for large vision models[C]//Proceedings of the IEEE/CVF Conference on Computer Vision and Pattern Recognition. 2024: 22861-22872.
> >
> > **Thanks again for spending your time on reading our response! We hope our answers are persuasive and have dispelled your doubts.**

---

> > > ### Author Response · Authors · 2024-12-01
> > > **Thanks to Reviewer rVKM!**
> > >
> > > Dear Reviewer rVKM,
> > >
> > > We sincerely appreciate your time and effort in reviewing our manuscript and offering valuable suggestions. As the author - reviewer discussion phase is drawing to a close, we would like to confirm whether our responses have effectively addressed your concerns.
> > >
> > > We provided detailed responses to your concerns a few days ago, and we hope they have adequately addressed your issues. If you require further clarification or have any additional concerns, please do not hesitate to contact us. We are more than willing to continue our communication with you.
> > >
> > > Best regards,
> > >
> > > The Authors

---

> > > > ### Comment · Reviewer_rVKM · 2024-12-01
> > > >
> > > > I have carefully reviewed the authors' reply and found that my concerns have been thoroughly addressed. I believe this is a well-written and valuable paper.

---

> ### Author Response · Authors · 2024-12-02
> **Thanks to Reviewer rVKM!**
>
> Dear Reviewer rVKM,
>
> We are truly overjoyed and deeply grateful to receive your kind and positive feedback. Your recognition of our efforts in addressing the concerns and the overall quality of the paper means a great deal to us. It is extremely encouraging and gives us a strong sense of accomplishment.
>
> We sincerely hope that you can raise the rating score, as it would have a significant impact on the future prospects and dissemination of our research.
>
> Best regards,
>
> The Authors

---

### Official Review · Reviewer_RLim · 2024-10-31

**Soundness:** 3
**Presentation:** 3
**Contribution:** 3
**Rating:** 6
**Confidence:** 4

**Summary:**

This paper presents a novel multimodal generative recommendation method, MQL4GRec, which facilitates the effective transfer of recommendation knowledge by converting item content from different domains and modalities into a unified quantitative language. Specifically, MQL4GRec introduces a unified quantitative language representation to handle multimodal content, including text and images. Additionally, a series of quantitative language generation tasks are designed to enrich the semantic representation. Extensive experiments across three datasets show that this method significantly outperforms baseline approaches.

**Strengths:**

* High Innovation: This work is the first to propose using a unified quantitative language to address knowledge transfer in multimodal recommendation, which is highly valuable for enhancing the generalization ability of recommendation systems.

* Thorough Experimental Validation: The paper conducts extensive experiments across three datasets, showcasing not only the overall performance of the method but also analyzing the role of individual components through ablation studies, indicating rigorous experimental design.

**Weaknesses:**

* The model's high complexity poses significant computational and storage demands, which could lead to considerable costs in real-world deployment.

**Questions:**

* Figure 3 shows that the performance on the Games dataset slightly declines as the amount of pre-training data increases. Does this suggest that the pre-training strategy has limitations when applied to domains with substantial cross-domain differences?

---

> ### Author Response · Authors · 2024-11-22
> **Rebuttal by Authors**
>
> **Thank you very much for your insightful comments on our work！ We try to answer your questions as follows.**
>
> >*Q1: The model's high complexity poses significant computational and storage demands, which could lead to considerable costs in real-world deployment.*
>
> A1: The method primarily consists of three models: two RQ-VAE models for tokenization and a transformer model for generative recommendations:
>
> - The RQ-VAE model includes an encoder, a decoder, and a quantization component. Both the encoder and decoder are implemented as Multi-Layer Perceptrons (MLPs) with ReLU activation functions. The encoder has six intermediate layers of sizes 2048, 1024, 512, 256, 128, and 64, culminating in a final latent representation dimension of 32. The time complexity of vector quantization is given by: $ O\left(\sum_{i=1}^k\left(n_{i-1} \times n_i\right)\right) $. Here, $ k $ represents the number of layers in the multi-layer perceptron, $ n_0 $ is the dimensionality of the item representation, and $ n_i $ is the number of nodes in the $ i $-th layer.
>   -  The specific training and inference times are related to the number of items. We set the batch size to 1024 and the maximum number of epochs to 500, and we use a Tesla V100 for training. Taking the Instruments dataset as an example, which has 6250 items, each epoch takes approximately 0.4 seconds, so training the vector quantization model takes about 4 minutes and occupies about 1.5GB of GPU memory. We perform one round of inference to obtain tokens for each item and save the distances from the residual vectors to the codebook, then handle item conflicts as described in the paper, a process that takes only about 1 second.
>   - After training the RQ-VAE model, we can use it as a general-purpose tokenizer, and the model has only about 20 million parameters.
>
> - We implement our transformer-based generative recommendation model using the T5 framework. Details of the model are introduced in Appendix C. This model has only about 8 million parameters.
>
> - **Most importantly, the size of our model does not increase with the number of items.** Traditional methods that rely on item IDs or item representations need to maintain an embedding for each item, and when the number of items is very large, the model parameters and storage space will increase dramatically. In contrast, our model has a fixed-size vocabulary, and we only need to store the tokens for each item. Compared to storing embeddings, this cost is negligible.
>
> - During the testing phase, MQL4GRec utilizes autoregressive decoding and beam search for inference. Its time complexity is given by $ O\left(KNL\left(d N^2+N d^2\right)\right) $, where $ L $ is the number of model layers, $ N $ is the sequence length, and $ d $ is the dimension of the hidden states, and $ K $ is the beam size. Additionally, a beam search with $ H $ steps introduces a time complexity of $ O(NKV\log K) $, with $ V $ denoting the vocabulary size. Traditional matrix factorization methods, which are the most efficient, require a time complexity of $ O(Id) $ to complete all inner product computations and $ O(I \log K) $ to identify the top-K recommendations for a single user, where $ I $ is the number of items. This suggests that generative recommendations may reduce inference time in cases where $ I \gg KNL\left(N^2+N d\right) > NKL $.
>
> >*Q2: Figure 3 shows that the performance on the Games dataset slightly declines as the amount of pre-training data increases. Does this suggest that the pre-training strategy has limitations when applied to domains with substantial cross-domain differences?*
>
> A2: Figure 3, when using both text and image quantitative languages for pre-training, the performance on the Games dataset decreases as the size of the pre-training dataset increases. However, when pre-trained only with text quantitative language ($NIG_1$ w/ pre-training), the performance increases. We believe there might be two potential reasons for this: 1) When pre-training with both text and image quantitative languages, the model may overfit; 2) The domain difference between the pre-training dataset and the Games dataset is significant, leading to insufficient generalization of the pre-trained model.
>
> **Thanks again for spending your time on reading our response! We hope our answers are persuasive and have dispelled your doubts.**

---

> > ### Author Response · Authors · 2024-12-01
> > **Thanks to Reviewer RLim!**
> >
> > Dear Reviewer  RLim,
> >
> > We sincerely appreciate your time and effort in reviewing our manuscript and offering valuable suggestions. As the author - reviewer discussion phase is drawing to a close, we would like to confirm whether our responses have effectively addressed your concerns.
> >
> > We provided detailed responses to your concerns a few days ago, and we hope they have adequately addressed your issues. If you require further clarification or have any additional concerns, please do not hesitate to contact us. We are more than willing to continue our communication with you.
> >
> > Best regards,
> >
> > The Authors

---

> > > ### Author Response · Authors · 2024-12-02
> > > **Looking forward to your reply!**
> > >
> > > Dear Reviewer RLim,
> > >
> > > As the author - reviewer discussion phase is drawing to a close, we would like to confirm whether our responses have effectively addressed your concerns. If we have resolved your issues, we sincerely hope that you can raise the rating score.
> > >
> > > Best regards,
> > >
> > > The Authors

---

### Official Review · Reviewer_vNzn · 2024-11-02

**Soundness:** 3
**Presentation:** 3
**Contribution:** 3
**Rating:** 6
**Confidence:** 4

**Summary:**

MQL4GRec introduces a novel approach for generative recommendation by transforming multimodal content from different domains into a unified "quantitative language," facilitating cross-domain knowledge transfer in recommendation tasks. The method uses quantitative translators for text and image content, building a shared vocabulary to encode semantic information across modalities. A series of language generation tasks further enriches this vocabulary, enhancing the model's capacity to represent multi-faceted user preferences. Experimental results demonstrate notable performance improvements over baseline models on key metrics across multiple datasets, showcasing MQL4GRec's scalability and potential in multimodal recommendation.



However, its innovation appears limited, closely resembling existing methods like GenRet and MMGRec in both tokenization approach and generative structure.

**Strengths:**

1. Innovation: The proposed MQL4GRec method translates content from different modalities into a unified “quantitative language,” enabling cross-domain and cross-modal recommendation knowledge transfer. This approach addresses limitations in handling multimodal data in existing generative recommendation models.


2. Superior Performance: Experimental results on multiple public datasets demonstrate that MQL4GRec outperforms baseline methods on key metrics such as NDCG.



3. Open-Source Availability: The paper provides a fully accessible code repository, facilitating reproducibility and further research within the community.

**Weaknesses:**

1. Similarity to Existing Tokenization Approaches
- The unified vocabulary for multimodal information closely resembles the generative retrieval tokenization approach found in "Learning to Tokenize for Generative Retrieval," [1]  particularly the "GenRet" model, which uses discrete auto-encoding for compact identifiers. This "multimodal codebook" seems to be an adaptation of single-modality tokenization, relying on established techniques like RQ-VAE and offering only incremental improvements without substantial architectural or performance innovation.
- The motivational structure and initial figures in MQL4GRec are closely aligned with those in GenRet, which may diminish the perceived originality of the proposed approach.



2. Lack of Comparative Analysis
-  MQL4GRec’s generative approach appears heavily inspired by MMGRec [2], which also employs Graph RQ-VAE for multimodal representation through user-item interactions, raising concerns about the uniqueness of MQL4GRec’s contributions.
- The paper does not clearly distinguish MQL4GRec's advancements over MMGRec, especially in terms of multimodal token-based representations. A more thorough comparison is needed to establish any unique contributions beyond MMGRec’s existing framework.


[1] Sun, W., Yan, L., Chen, Z., Wang, S., Zhu, H., Ren, P., ... & Ren, Z. (2024). Learning to tokenize for generative retrieval. Advances in Neural Information Processing Systems, 36.

[2 ] Liu, H., Wei, Y., Song, X., Guan, W., Li, Y. F., & Nie, L. (2024). MMGRec: Multimodal Generative Recommendation with Transformer Model. arXiv preprint arXiv:2404.16555.

The paper does not clearly distinguish MQL4GRec's advancements over MMGRec, especially in terms of multimodal token-based representations. A more thorough comparison is needed to establish any unique contributions beyond MMGRec’s existing framework.

**Questions:**

- User ID Collision: With only 2000 tokens representing a large user base, there is a potential for ID collisions, which may lead to inaccuracies in recommendation results in real-world applications.
   - Domain Adaptability: The model performs poorly on certain domain-specific datasets, such as the Games dataset, suggesting limitations in domain transferability.

---

> ### Author Response · Authors · 2024-11-22
> **Rebuttal by Authors**
>
> **Thank you very much for your insightful comments on our work！ We try to answer your questions as follows.**
>
> >*Q1: Similarity to Existing Tokenization Approaches.*
>
> A1: Our tokenization approach indeed got inspiration from the well established RQ-VAE technology as other excellent and innovative works do, such as TIGER [1], LC-Rec [2] and LETTER [3]. We believe that our approach's innovation and originality lie in its unique application and integration of RQ-VAE within our model's broader framework, which distinguishes it from existing work. We would like to list some key points as follows.
>
> - Unlike previous methods, we propose a new approach to handle collisions. Previous methods are all aimed at a single dataset and can not address the issue of large-scale item collisions across multi-domain datasets, and they would introduce semantically unrelated distributions. Our proposed method for handling collisions can perfectly solve the above problems.
>
> - Moreover, the core innovation of our method lies in transforming the content of items from different domains and modalities into a unified quantitative language, thereby breaking down the barriers between them. This novel idea is precisely achieved through multimodal tokenization.
>
> - After obtaining the multimodal quantitative language, we design a series of quantitative language generation tasks, and implement the transfer of recommendation knowledge across different domains and modalities through pre-training and fine-tuning. This method possesses novelty and originality.
>
> Lastly, could it be that there was a typographical error in this section of the review, where MMGRec was mistakenly referred to as GenRet, given that GenRet does not incorporate multimodal information and RQ-VAE?
>
> [1] Rajput S, Mehta N, Singh A, et al. Recommender systems with generative retrieval[J]. Advances in Neural Information Processing Systems, 2023, 36: 10299-10315.
>
> [2] Zheng B, Hou Y, Lu H, et al. Adapting large language models by integrating collaborative semantics for recommendation[C]//2024 IEEE 40th International Conference on Data Engineering (ICDE). IEEE, 2024: 1435-1448.
>
> [3] Wang W, Bao H, Lin X, et al. Learnable Item Tokenization for Generative Recommendation[C]//Proceedings of the 33rd ACM International Conference on Information and Knowledge Management. 2024: 2400-2409.

---

> > ### Author Response · Authors · 2024-11-22
> > **Rebuttal by Authors**
> >
> > >*Q2: Lack of Comparative Analysis.*
> >
> > A2: While both our method and MMGRec utilize multimodal information of items, the two approaches are fundamentally different:
> >
> > - First, **our recommendation tasks are distinct**.
> >   - Our method is designed for sequential recommendation tasks, whereas MMGRec is not.  As mentioned in the Baselines section of the MMGRec paper, "Note that we did not select the existing generative recommendation model TIGER as a baseline, since it only works with sequential recommendation rather than our task."
> >
> > - Secondly, **our motivations are different**.
> >   - The core idea of MMGRec is to learn multimodal representations of items to enhance generative recommendations. However, our core idea is to transform the recommendation task into a special type of language generation task, thereby facilitating the transfer of recommendation knowledge across different domains and modalities. This concept is inspired by the significant successes of recent advancements in natural language generation. We aspire to emulate this success by translating the content of items from various domains and modalities into a quantitative language, and by designing diverse generation tasks to imbue the model with rich semantic information and enable the transfer of recommendation knowledge.
> >
> > - Thirdly, **our contributions are distinctly different**.
> >   - MMGRec designs the Graph RQ-VAE and proposes a relation-aware self-attention mechanism.
> >   - We propose transforming the content of items from different domains and modalities into a unified quantitative language and then design a series of quantitative language generation tasks.
> >
> > - Fourthly, **our methods are different**.
> >   - During the tokenization phase, MMGRec designs the Graph RQ-VAE and utilizes multimodal and collaborative filtering (CF) information for Rec-ID assignment, then addresses item collisions based on item popularity. To verify whether this tokenization method is more effective, we compare it with the $NIG_1$ task. As can be seen from the table below, despite Graph RQ-VAE utilizing more information, its performance is not particularly good. This may be because Graph RQ-VAE has to balance Rec-ID assignment with learning the multimodal representations of items.
> >   - Our approach to utilizing multimodal information is different. MMGRec uses Graph RQ-VAE for Rec-ID assignment and simultaneously learns the multimodal representations of items, then uses the item representations as input embeddings for the transformer model. Although this method has achieved good results in MMGRec, it is essentially still limited to learning item representations. This approach is fundamentally different from our core idea:
> >     - Our goal is not to use multimodal representations of items to improve recommendation performance, but rather to transform the different modalities of item content into a unified linguistic form, and then use language generation tasks to facilitate the transfer of recommendation knowledge across different domains and modalities.
> >   - MMGRec designs a relation-aware self-attention mechanism for the Transformer to handle non-sequential interaction sequences, thereby enhancing recommendation performance. In contrast, we design a series of quantitative language generation tasks that endow quantitative language with rich semantic information and prior knowledge, and enhance the performance of recommendation tasks through pre-training and fine-tuning.
> >
> > - **Our method possesses greater flexibility and scalability.**
> >   - MMGRec utilizes not only multimodal information but also collaborative filtering (CF) information during tokenization. Due to the lack of CF information for new items, it may learn inferior item representations and Rec-IDs. Our method does not have this drawback.
> >   - MMGRec struggles to scale to large-scale datasets and consumes more storage space. The model input for MMGRec is still limited to item IDs and item representations, with the number of embeddings at the model input equal to the number of items. When the number of items reaches millions or even billions, the model parameters will increase dramatically. Our method's input consists of task prompt tokens and item tokens, with a fixed vocabulary size, so the model parameters do not increase with the number of items. Assuming each item token has a quantity of 4, we can represent 100 million items with 400 (100x4) tokens.
> >   - MMGRec is limited to item IDs and item representations and cannot achieve knowledge transfer across multiple domains. Our method transforms the content of items from different domains and modalities into a unified linguistic form, thereby enabling flexible knowledge transfer.

---

> > > ### Author Response · Authors · 2024-11-22
> > > **Rebuttal by Authors**
> > >
> > > |                | Instruments  |         |        |        |         | Arts   |         |        |        |         | Games   |         |        |        |         |
> > > | ---------      | ------       | ------- | ------ | ------ |---------| ------ | ------- | ------ | ------ | ------  | ------  | ------- | ------ | ------ | ------  |
> > > |                | HR@1         | HR@5    | HR@10  | NDCG@5 | NDCG@10 | HR@1   | HR@5    | HR@10  | NDCG@5 | NDCG@10 |  HR@1   | HR@5    | HR@10  | NDCG@5 | NDCG@10 |
> > > | Graph RQ-VAE   | 0.0680       | 0.0917  | 0.1109 | 0.0801 |  0.0863 | 0.0565 | 0.0797  | 0.0937 | 0.0681 |  0.0727 |  0.0166 | 0.0462  | 0.0752 | 0.0313 |  0.0406 |
> > > | $NIG_1$        | **0.0780**       | **0.1034**  | **0.1277** | **0.0909** |  **0.0987** | **0.0604** | **0.0931**  | **0.1163** | **0.0769** |  **0.0844** |  **0.0183** | **0.0558**  | **0.0885** | **0.0368** |  **0.0473** |
> > >
> > > >*Q3: User ID Collision: With only 2000 tokens representing a large user base, there is a potential for ID collisions, which may lead to inaccuracies in recommendation results in real-world applications.*
> > >
> > > A3: TIGER takes user IDs as input for the model, but we do not use user IDs because users across different domains are non-overlapping. We only model the historical interaction sequences of users. Table 2 is provided for a fair comparison to demonstrate the effectiveness of our proposed method for handling collisions.
> > >
> > >
> > > >*Q4: Domain Adaptability: The model performs poorly on certain domain-specific datasets, such as the Games dataset, suggesting limitations in domain transferability.*
> > >
> > > A4: We discuss these experimental results in section 4.4. When using both text and image quantitative languages for pre-training, the performance on the Games dataset decreases as the size of the pre-training dataset increases. However, when pre-trained only with text quantitative language ($NIG_1$ w/ pre-training), the performance increases. We believe there might be two potential reasons for this: 1) When pre-training with both text and image quantitative languages, the model may overfit; 2) The domain difference between the pre-training dataset and the Games dataset is significant, leading to insufficient generalization of the pre-trained model.
> > >
> > > **Thanks again for spending your time on reading our response! We hope our answers are persuasive and have dispelled your doubts. We sincerely hope that you can give a high score.**

---

> > > > ### Author Response · Authors · 2024-12-01
> > > > **Thanks to Reviewer vNzn!**
> > > >
> > > > Dear Reviewer vNzn,
> > > >
> > > > We sincerely appreciate your time and effort in reviewing our manuscript and offering valuable suggestions. As the author - reviewer discussion phase is drawing to a close, we would like to confirm whether our responses have effectively addressed your concerns.
> > > >
> > > > We provided detailed responses to your concerns a few days ago, and we hope they have adequately addressed your issues. If you require further clarification or have any additional concerns, please do not hesitate to contact us. We are more than willing to continue our communication with you.
> > > >
> > > > Best regards,
> > > >
> > > > The Authors

---

### Official Review · Reviewer_AB8j · 2024-11-03

**Soundness:** 3
**Presentation:** 4
**Contribution:** 4
**Rating:** 6
**Confidence:** 4

**Summary:**

The paper titled "Multimodal Quantitative Language for Generative Recommendation" introduces a novel approach to enhance generative recommendation systems by converting item content from multiple domains and modalities into a unified "quantitative language". This methodology seeks to bridge the gap between the generalized linguistic knowledge of pre-trained language models (PLMs) and the specialized needs of recommendation systems. The authors developed a new framework, MQL4GRec, which employs "quantitative translators" to convert textual and visual item data into a shared vocabulary. This shared language is then enriched with semantic information through various generation tasks to enable effective knowledge transfer from multimodal data to recommendation systems.

The paper's main contribution lies in its innovative method of integrating multimodal data to improve recommendation performance significantly, surpassing baseline methods by notable margins in terms of the NDCG metric across multiple datasets. Furthermore, the framework introduces a potential shift towards more universal recommendation systems that do not rely on traditional item IDs, thereby addressing common challenges like improving scalability and transferability across different domains.

**Strengths:**

1. The approach to transform diverse item content from different domains and modalities into a unified quantitative language is highly innovative. This integration allows for a more robust and versatile recommendation system that can handle varied inputs effectively.

2. The paper introduces a well-structured design of pre-training and fine-tuning tasks that includes not only generative prediction but also alignment tasks, enhancing the robustness of the method. This comprehensive approach allows the model to effectively leverage both generative capabilities and alignment strategies to improve overall recommendation accuracy.

3. The proposed framework significantly outperforms existing models on several benchmark datasets, particularly in terms of the NDCG metric. This suggests that the method is not only theoretically sound but also practically effective.

**Weaknesses:**

1. The paper does not discuss the impact of various training operations on the algorithm's time complexity, such as the handling of collisions. This oversight might leave questions about the scalability and efficiency of the proposed method in practical applications.

2. The methodology section lacks a clear explanation of how the generated quantitative vectors are used to retrieve corresponding items during the next item generation task. This omission could lead to ambiguity regarding the operational specifics of the model.

3. The model does not utilize multimodal information simultaneously to predict the next click item, which limits the paper's innovativeness and potential for expansion. Leveraging such multimodal data more effectively could enhance the model's predictive performance and applicability in more complex scenarios.

**Questions:**

1. How do different training operations, such as collision handling and quantization, impact the computational time and resource requirements of the model?

2. What methods or algorithms are employed to map these quantitative vectors back to specific items in the dataset?

3. Why does the model not use multimodal information simultaneously to predict the next click item? What are the challenges or limitations that prevent this integration?

---

> ### Author Response · Authors · 2024-11-22
> **Rebuttal by Authors**
>
> **Thank you very much for your insightful comments on our work！ We try to answer your questions as follows.**
>
> >*Q1: How do different training operations, such as collision handling and quantization, impact the computational time and resource requirements of the model?*
>
> A1: Our proposed MQL4GRec consists of four distinct steps: vector quantization, collision resolution, training of the recommendation model, and testing. Below, we will provide a detailed introduction:
>
> - Firstly, we train RQ-VAE models using item representations extracted from pre-trained modal encoders. Both the encoder and decoder of the RQ-VAE are implemented as Multi-Layer Perceptrons (MLPs) with ReLU activation functions. The encoder has six intermediate layers of size 2048, 1024, 512, 256, 128 and 64 with a final latent representation dimension of 32. The time complexity of vector quantization is: $O\left(\sum_{i=1}^k\left(n_{i-1} \times n_i\right)\right)$, where $k$ is the number of layers in the MLP, $n_0$ is the dimension of the item representation, and $n_i$ is the number of nodes in the $i$-th layer.
>
>   - Specifically, the training time is dependent on the number of items. We set the batch size to 1024 and the maximum epochs to 500, and use a Tesla V100 for training. For example, the Instruments dataset contains 6250 items, with each epoch taking about 0.4 seconds. Training the vector quantization model takes approximately 4 minutes and occupies about 1.5GB of VRAM.
>
> - After training the RQ-VAE, we assign tokens to each item and resolve item collisions. We perform one inference pass to obtain the tokens for each item, and save the distances from the residual vectors to the codebook. We then handle item collisions using the method described in the paper.
>   - Taking the Instruments dataset as an example, this process requires only about 1 second.
>
> - After assigning tokens to each item, we carry out the pre-training and fine-tuning of the recommendation model, the details of which can be found in Appendix C. The time complexity of this model is $ O\left(L\left(d N^2 + N d^2\right)\right) $, where $ L $ is the number of model layers, $ N $ is the sequence length, and $ d $ is the dimension of the hidden states.
>
>   - Vector quantization has only one parameter that affects the time complexity of the recommendation model, which is the number of levels in the vector quantizer. This determines the number of tokens assigned to each item. The more tokens per item, the longer the input and output sequences will be, leading to higher time complexity and resource requirements.
>   - Collision resolution has no impact on the time complexity of the recommendation model.
>
> - During the testing phase, MQL4GRec employs autoregressive decoding and beam search for inference. Its time complexity is given by: $ O\left(KNL\left(d N^2 + N d^2\right)\right) $, where $ K $ is the beam size. Additionally, a beam search with $ H $ steps introduces a time complexity of $ O(NKV\log K) $, with $ V $ denoting the vocabulary size.
>   - Similarly, the number of levels in the vector quantizer will affect the inference time and resource requirements, while collision resolution does not have any impact.
>
>
> >*Q2: What methods or algorithms are employed to map these quantitative vectors back to specific items in the dataset?*
>
> A2: The purpose of vector quantization and collision resolution is to assign a unique set of tokens to each item, where the codebook vectors and residual vectors involved are only used to calculate the distances between vectors and are not used in the subsequent recommendation model.
> - After assigning tokens to each item, we maintain dictionaries from items to tokens and from tokens to items.
> - When training the transformer model, the embeddings corresponding to the vocabulary are randomly initialized, rather than being initialized with quantitative vectors.
>   - This is because the quantitative vectors mainly contain semantic information of the items, and we hope to automatically learn word vectors suitable for recommendation tasks through various generation tasks.
> - During transformer decoding, the last layer's hidden state will pass through a linear layer and a softmax layer to obtain the token ID. By looking up the vocabulary, we can get the decoded tokens. Finally, we look up the dictionary mapping from tokens to items.

---

> ### Author Response · Authors · 2024-11-22
> **Rebuttal by Authors**
>
> >*Q3: Why does the model not use multimodal information simultaneously to predict the next click item? What are the challenges or limitations that prevent this integration?*
>
> A3: Thanks for this great suggestion, and in the future, we will also explore a better way to simultaneously and effectively utilize multimodal information. Actually we use multimodal data for some specific tasks and combine the result together for item retrieval. We have to mention that using multimodal data in a more compact way may face some challenges.
>
> - Joining the text tokens and visual tokens as a single tuple increases the length of representing an item, which in turn raise the complexity of the generation tasks and lead to a degradation in efficiency.
> - There is a necessity to devise a more diverse array of generation tasks. While this enhances the variety of training samples, it also introduce the potential issue of overfitting.
>
> However, exploring a better way of leveraging the multimodal information as suggested is a valuable direction and we will consider as our future work.
>
>
> **Thanks again for spending your time on reading our response! We hope our answers are persuasive and have dispelled your doubts.**

---

> ### Author Response · Authors · 2024-12-01
> **Thanks to Reviewer AB8j!**
>
> Dear Reviewer  AB8j,
>
> We sincerely appreciate your time and effort in reviewing our manuscript and offering valuable suggestions. As the author - reviewer discussion phase is drawing to a close, we would like to confirm whether our responses have effectively addressed your concerns.
>
> We provided detailed responses to your concerns a few days ago, and we hope they have adequately addressed your issues. If you require further clarification or have any additional concerns, please do not hesitate to contact us. We are more than willing to continue our communication with you.
>
> Best regards,
>
> The Authors

---

> > ### Author Response · Authors · 2024-12-02
> > **Looking forward to your reply!**
> >
> > Dear Reviewer AB8j,
> >
> > As the author - reviewer discussion phase is drawing to a close, we would like to confirm whether our responses have effectively addressed your concerns. If we have resolved your issues, we sincerely hope that you can raise the rating score.
> >
> > Best regards,
> >
> > The Authors

---

### Author Response · Authors · 2024-11-22
**Rebuttal by Authors**

We thank all the reviewers for their time and effort in reviewing our paper and providing constructive feedback！

We are encouraged that **all reviewers have recognized the high novelty and innovation** of our proposed MQL4GRec, as well as the extensive experimental validation that demonstrates the effectiveness of our approach. We are grateful for the high praise from **all reviewers** regarding the openness of our anonymized code repository, which greatly enhances the reproducibility of our work.

Additionally, we sincerely appreciate the reviewers' appreciation for our rigorous experimental design, including comprehensive experimental validation and ablation studies (Reviewer **RLim**), the elegance of our architecture (Reviewer **rVKM**), and the well-structured design of pre-training and fine-tuning tasks (Reviewer **AB8j**). We are especially thankful to the fourth reviewer for appreciating this type of work and believing that it can make a meaningful impact in the field of generative recommendation (Reviewer **rVKM**).

After carefully analyzing the reviewers' comments, we are more than glad to find these comments very insightful. We respond to the comments point by point in the rebuttal. We hope that our response can address the concerns raised by the reviewers.

Once again, we sincerely thank the reviewers for their valuable feedback and insightful suggestions, which undoubtedly contribute to enhancing the quality of our work.

---

### Meta-Review · Area_Chair_Pn6z · 2024-12-24

**Metareview:**

This paper introduces a novel approach called Multimodal Quantitative Language for Generative Recommendation (MQL4GRec), which transforms items from different domains and modalities into a unified language and transfers recommendation knowledge across different domains. The authors have performed extensive experiments on three different datasets to demonstrate the effectiveness of the proposed method.

Overall, this paper is clearly written. The ideas of the proposed method are novel, e.g., transforming diverse item content from different domains and modalities into a unified quantitative language is novel, and the well-structured design of pre-training and fine-tuning tasks. The experimental evaluation is extensive and convincing. The proposed method outperforms existing methods on benchmark datasets.

**Additional Comments On Reviewer Discussion:**

In the rebuttal, the authors provide more discussions about the impact of different training operations, the mapping of quantitative vectors to specific items, the similarity between the proposed method and existing tokenization methods, and the complexity of the proposed method. Moreover, they also provide the statistical significance analysis of MQL4GRec results, some experimental results about the training costs of the proposed method. Some of the reviewers' concerns have been well addressed.

---

### Decision · Program_Chairs · 2025-01-22

Accept (Poster)